# Intra-Arterial Urokinase for Acute Superior Mesenteric Artery Occlusion: A Retrospective 12-Year Report of 13 Cases

**DOI:** 10.3390/biomedicines11020267

**Published:** 2023-01-18

**Authors:** Being-Chuan Lin, Cheng-Hsien Wu, Yon-Cheong Wong, Sheng-Che Hung, Ming-Che Hsin

**Affiliations:** 1Division of Trauma & Emergency Surgery, Department of Surgery, Chang Gung Memorial Hospital, Chang Gung University, Tao-Yuan City 33302, Taiwan; 2Division of Emergency and Critical Care Radiology, Department of Medical Imaging and Intervention, Chang Gung Memorial Hospital, Chang Gung University, Tao-Yuan City 33302, Taiwan; 3Department of Radiology and BRIC, University of North Carolina at Chapel Hill, Chapel Hill, NC 27599, USA; 4Body Science & Metabolic Disorders International (BMI) Medical Center, China Medical University Hospital, Taichung City 404327, Taiwan

**Keywords:** intra-arterial urokinase, superior mesenteric artery occlusion, thrombolysis, recanalisation degree, bowel perfusion

## Abstract

This retrospective study aimed to evaluate the outcomes of 13 patients with acute superior mesenteric artery (SMA) occlusion who underwent intra-arterial urokinase thrombolysis between 2008 and 2020. On angiography, seven presented with complete SMA occlusion versus six with incomplete occlusion. The median time from abdominal pain to attempting urokinase thrombolysis was 15.0 h (interquartile range, 6.0 h). After urokinase therapy, bowel perfusion was restored with bowel preservation in six patients; however, treatment failed in the other seven patients. The degree of SMA occlusion (complete vs. incomplete, *p* = 0.002), degree of recanalisation (*p* = 0.012), and length of stay (*p* = 0.032) differed significantly between groups. Of the seven patients with complete SMA occlusion, six underwent bowel resection, of whom three died, and the remaining patient died of shock due to delayed surgery. Among the six patients with incomplete SMA occlusion, no bowel resection was performed. In our experience, intra-arterial urokinase thrombolysis may serve as an adjunctive treatment modality, being a potential replacement for open thrombectomy that is able to preserve the bowel and obviate surgery in cases of incomplete SMA occlusion; however, its use is unsuitable in cases of complete SMA occlusion, for which surgery is warranted.

## 1. Introduction

Acute mesenteric ischaemia (AMI) accounts for 1–2% of acute abdominal emergencies [1,2]. If left untreated, AMI causes bowel infarction, necrosis, sepsis and death. Despite progress in diagnostic and treatment strategies for this vascular emergency, it remains life-threatening with an overall mortality rate of 50–70% [3,4,5]. Superior mesenteric artery (SMA) occlusion by embolism (50%) and thrombosis (15–25%) are the most common causes of AMI. To restore SMA blood perfusion and preserve the bowel, intra-arterial thrombolysis may be used as an adjunctive treatment to surgery. However, only sporadic cases have been reported; even in large case series, few cases exist [6,7,8,9,10]. This study aimed to present our experience using intra-arterial urokinase thrombolysis in 13 patients with acute SMA occlusion and analyse their outcomes.

## 2. Materials and Methods

### 2.1. Patient Selection

This retrospective study was approved by the Institutional Review Board of Chang Gung Memorial Hospital (202000381B0) on 13 March 2020, which waived the requirement for informed consent due to our retrospective analysis of anonymised data. The case series was structured in accordance with the PROCESS criteria [11]. Patients who underwent intra-arterial urokinase thrombolysis for acute SMA occlusion between 1 October 2008 and 15 July 2020 were identified in the registry.

### 2.2. Indication of Thrombolysis

Patients who presented with sudden abdominal pain but an unrevealing abdominal examination, nausea, vomiting, and/or rectal bleeding were highly suspected to have ischaemic bowel and diagnosed with acute SMA occlusion on contrast-enhanced abdominal computed tomography (CT).

### 2.3. Exclusion Criteria

Patients presenting with shock in the triage screening, acute peritonitis on a physical examination, intramural gas, and mesenteric or portal venous gas on contrast-enhanced abdominal CT were excluded from treatment. During the study period, 14 patients were diagnosed with acute SMA occlusion on contrast-enhanced abdominal CT and scheduled for intra-arterial urokinase thrombolysis. One patient who underwent intra-arterial fragmentation alone due to mesenteric contrast extravasation on angiography was excluded from the analysis. The remaining 13 patients, who received intra-arterial urokinase infusions, were selected for this study.

### 2.4. Procedure

The thrombolysis procedures were performed by interventional radiologists. Under local anaesthesia, the right femoral artery was punctured in accordance with the Seldinger technique, and a 10 cm long 6 Fr sheath (Terumo, Tokyo, Japan) was implanted. Selective catheterisation of the SMA was performed using an 80 cm long 4 Fr catheter (J curve; Terumo). SMA angiography was performed to identify the filling defects. Thrombolysis was performed using a 5 Fr multiple-sideport infusion catheter (100 cm with fourteen 7 cm sideports or 100 cm with thirty 15 cm sideports; Cook, Bloomington, IN, USA). The tip of the microcatheter was embedded in the thromboembolism, and our infusion protocol was started with intrathrombus pulse-spray injection of urokinase (urokinase-GCC injection, 250,000 IU) with a loading dose of 300,000 IU in 20 mL of normal saline in the first 3 patients and 250,000 IU in the next 10 patients, followed by a maintenance dose of 50,000 IU/h for 3 days. Intravenous heparin was administered simultaneously under close monitoring and surveillance in the surgical intensive care unit. Possible hemorrhagic complications, such as intracranial haemorrhage, gastrointestinal bleeding, or puncture site oozing, were assessed, and the fibrinogen levels were checked every 6 h and the dose of urokinase was adjusted or discontinued for fibrinogen < 200 mg/dL. Follow-up angiography was usually performed once daily for 3 days or discontinued when clinical deterioration occurred. The patient was discharged with a warfarin prescription. Data on age, sex, clinical presentation, imaging studies such as abdominal CT and angiography, SMA occlusion location and degree, time and response to urokinase treatment, and clinical outcomes were retrospectively evaluated.

### 2.5. Definition

SMA occlusions were classified into proximal and distal occlusions defined as thromboembolisms proximal and distal to the middle colic artery, respectively. SMA occlusion degree was defined as complete (Figure 1A,B) or incomplete (Figure 2A,B) for a main trunk of SMA occlusion with and without distal branches, respectively. Recanalisation degree after intra-arterial urokinase thrombolysis was described as total, near-total (Figure 3A,B), partial (Figure 3C,D), or absent for total, near-total, partial, and no restoration of blood flow on angiography, respectively.

### 2.6. Statistical Analysis

Categorical data are presented as numbers, whereas continuous data are presented as medians (interquartile range (IQR)). For comparisons of categorical data, Fisher’s exact test or the Pearson *χ*^2^ test was used as appropriate. The Mann–Whitney *U* test was used to examine continuous data. All statistical analyses were performed using SPSS version 20.0 (IBM, Armonk, NY, USA). Statistical significance was set at *p* < 0.05 (two-sided).

## 3. Results

Among the 13 patients, 9 were men and 4 were women (median age, 73.0 years). The median time from the onset of abdominal pain to emergency department admission was 9.0 h (IQR, 7.5 h). All had various degrees of acute abdomen, but none presented with acute peritonitis on a physical examination, and five (38.5%) had bright red blood per rectum. A medical history review revealed that hypertension (84.6%) and atrial fibrillation (AF; 69.2%) were the most common comorbidities. Laboratory data revealed an initial median serum white cell blood count of 13,200 U/L (IQR, 8350.0 U/L), median haemoglobin level of 14.0 g/dL (IQR, 4.5 g/dL), and median serum amylase level of 121.0 U/L (IQR, 79.0 U/L; Table 1). In this study, all 13 patients underwent contrast-enhanced abdominal CT for the diagnosis of SMA occlusion, which revealed additional renal ischaemia in 1 patient and synchronous spleen and renal ischaemia in another. On angiography, SMA occlusion was located proximally in four patients and distally in nine patients. Complete and incomplete SMA occlusions were observed in seven and six patients, respectively (Table 1). The median time from the onset of abdominal pain to the initiation of urokinase infusion was 15.0 h (IQR, 6.0 h; Table 1).

At the initiation of urokinase infusion and fragmentation for thromboembolism, early partial recanalisation was observed on angiography in six patients. After the urokinase therapy, all patients showed various degrees of recanalisation (near-total, *n* = 5; partial, *n* = 8) on follow-up angiography. However, owing to clinical deterioration (aggravated abdominal pain, *n* = 5; bright red blood per rectum, *n* = 2; shock, *n* = 1), urokinase therapy was discontinued in 8 patients. The median duration of urokinase therapy was 42.0 h (IQR, 45.0 h; Table 1). Details of the demographic characteristics, responses to urokinase therapy, and outcomes of the 13 patients are presented in Table 2. After urokinase therapy, bowel perfusion was restored with bowel preservation in six patients; however, in the other seven patients, bowel perfusion was not restored (Table 2). A comparison of the demographic data and clinical characteristics of the patients with restored and unrestored bowel perfusion revealed no differences in time from abdominal pain to initiation of urokinase administration (*p* = 0.281) and occlusion location (*p* = 0.105), while the degree of SMA occlusion (complete vs. incomplete, *p* = 0.002), degree of recanalisation (*p* = 0.012), and length of stay (*p* = 0.032) did differ (Table 3). Of the seven patients with complete SMA occlusion, six underwent bowel resection, of whom three died, while the remaining non-surgically treated patient died of shock due to surgical delay. Of the six patients with incomplete SMA occlusions, no bowel resection was performed, except for a partial omentectomy. Of the four deaths, one was attributed to surgical delay and the other three patients developed short bowel syndrome with sepsis and multiple organ failure, with a 30.8% in-hospital mortality rate. The median hospital stay duration was 17.0 days (IQR, 34.0 days) (Table 1).

## 4. Discussion

The high morbidity and mortality rates of AMI are mainly attributed to late diagnosis and treatment in elderly and debilitated cardiac patients, who are poor candidates for surgery [4,5,12,13]. The median patient age was 73.0 years (range, 50–88 years). All patients had cardiovascular disease (hypertension, *n* = 11; AF, *n* = 9; coronary artery disease, *n* = 4; congestive heart failure, *n* = 2; Table 2). To date, specific markers for establishing or excluding the diagnosis of SMA occlusion are lacking [4,5,14,15]. Laboratory data, such as leucocytosis, metabolic acidosis, hyperamylasaemia, and elevated lactate phosphate, are non-specific and suggestive of intestinal ischaemia. The most important diagnostic tool is contrast-enhanced abdominal CT, which not only facilitates the diagnosis of SMA occlusion, but shows other indications of ischaemic bowel, such as bowel wall thickening, bowel dilatation, intramural gas, mesenteric or portal venous gas, lack of bowel wall enhancement, and ischaemia of other abdominal organs [4,5,16,17,18]. In the past 20 years, many reports have described cases of successful reperfusion of SMA occlusion using several endovascular strategies, such as percutaneous intra-arterial thrombolysis, aspiration embolectomy, percutaneous transluminal angioplasty, SMA stenting, and a combination thereof [1,10,19,20]. The first case of intra-arterial thrombolysis for acute SMA occlusion was reported by Jamieson in 1979, in which streptokinase was successfully infused directly into the SMA [21].

Our first patient was a 50-year-old man who presented with abdominal pain. Contrast-enhanced abdominal CT revealed a non-enhanced filling defect in the distal part of the SMA on 1 October 2008. SMA angiography revealed incomplete SMA occlusion 5 cm distal to the SMA root (Figure 4A). Intra-arterial urokinase was administered at a loading dose of 300,000 IU, followed by a continuous dose of 50,000 IU/h. Follow-up angiography performed the next day revealed partial recanalisation of the previous occlusion with the appearance of distal branches (Figure 4B). However, the abdominal pain was aggravated, and emergency surgery was performed. During laparotomy, the small bowel and colon were normal with good perfusion, and only a fibrotic omental mass measuring 5 × 5 cm was found. The infarcted omentum was resected and the patient discharged uneventfully. We chose urokinase as the thrombolytic agent because it does not influence the surgery owing to its short half-life of only 16 min. In a literature review conducted from 1966 to 2003, Schoots et al. [9] reported that 48 patients with acute SMA thromboembolism were managed with intra-arterial thrombolysis, while 38 (79%) were treated with urokinase therapy. For our patients, intra-arterial fragmentation of thromboembolism was performed at the initiation of urokinase administration, but only six patients showed early partial recanalisation on angiography. Our experience revealed that the role of intra-arterial fragmentation in thromboembolism is limited. All of our patients achieved various degrees of SMA recanalisation on follow-up angiography (near-total, *n* = 5; partial, *n* = 8; Table 2). However, restoration of adequate bowel perfusion with bowel preservation was achieved in six, while restoration of bowel perfusion failed in the other seven. Of the 48 patients in the Schoots case series [9], excluding 3 in whom the technical approach failed, bowel preservation was achieved in 39 (86.7%). This revealed that, under thrombolysis for SMA occlusion, bowel perfusion restoration with bowel preservation was not clearly correlated in all patients and may be affected by many factors, such as occlusion duration and thrombolysis timing, degree of occlusion, presence or absence of collateral circulation, influence of splanchnic autoregulation, and presence of associated atherosclerotic lesions [4,5,22]. In our study, the time from abdominal pain to the initiation of urokinase administration (*p* = 0.281) and occlusion location (proximal vs. distal, *p* = 0.105) were not associated with bowel preservation (Table 3). However, degree of occlusion (complete vs. incomplete, *p* = 0.002) and recanalisation (near-total vs. partial, *p* = 0.012) were associated with bowel preservation (Table 3). Previous studies suggested initiating thrombolysis within 8–10 h after the appearance of abdominal symptoms [23]. In our patients, the median time from the onset of abdominal pain to the initiation of urokinase administration in six patients with restored bowel perfusion and in seven patients with unrestored bowel perfusion was 14.5 and 16.0 h, respectively, and this difference was not significant (*p* = 0.281). In addition, in Schoots’ case series, among 9 of 48 patients, 7 (14.6%) demonstrated thrombolysis success even at ≥24 h after presentation (range, 24–72 h) before angiography, and 4 (8.3%) failed thrombolysis even at ≤6 h before angiography. This discrepancy may suggest a longer window of opportunity for initiating thrombolysis in acute SMA occlusions.

Acute thrombosis of the SMA is generally localised in stenosis at the origin of the SMA, which is often a result of chronic atherosclerosis. Emboli are typically dislodged at the periphery of the SMA in atherosclerosis-free segments. Of the 13 patients, 3 presented with a proximal location and all had complete occlusion, for which only partial recanalisation was achieved after urokinase therapy (Table 2), which may reflect the reduced effect of urokinase on chronic atherosclerosis. Our report revealed, however, that the most important factor for bowel preservation was occlusion degree (complete versus incomplete, *p* = 0.002; Table 3). In our study, six patients had incomplete SMA occlusion, of whom five achieved near-total recanalisation after urokinase therapy, and all recovered well without bowel resection, except for the omentectomy performed in one patient (Table 2). Conversely, only partial recanalisation was achieved in the seven patients with complete occlusion, and six required bowel resection, of whom five underwent repeated surgery for extensive bowel resection and three died of short bowel syndrome with sepsis and multiple organ failure (Table 2). The surgery was delayed in an 83-year-old woman who presented with an acute abdomen, in whom angiography revealed a complete SMA occlusion 3.5 cm distal to the SMA root and intra-arterial urokinase was administered (patient 4 in Table 2). With urokinase therapy, her haemodynamic status deteriorated progressively, and surgery was scheduled; however, bradycardia soon developed, and she died without surgery. A laparoscopic examination was performed in four patients with equivocal clinical presentation, of whom two showed a normal bowel not requiring further laparotomy. Our experience in the management of acute SMA occlusion with intra-arterial urokinase revealed that it may serve as an adjunctive treatment modality, being a potential replacement for open thrombectomy that can preserve the bowel and obviate surgery in incomplete SMA occlusion; however, it was unsuitable for complete SMA occlusion. However, catheter-directed thrombolysis is not a covered service and therefore may be unavailable in your setting. Limitations: This was a retrospective 12-year study of 13 cases treated with the same technique in one centre. However, the small sample size prevented an appropriate multivariable analysis.

## 5. Conclusions

In our experience, intra-arterial urokinase thrombolysis may serve as an adjunctive treatment modality and be a potential replacement for open thrombectomy and preserve the bowel and obviate surgery in cases of incomplete SMA occlusion; however, its use is unsuitable in cases of complete SMA occlusion, for which surgery is warranted.

## Figures and Tables

**Figure 1 biomedicines-11-00267-f001:**
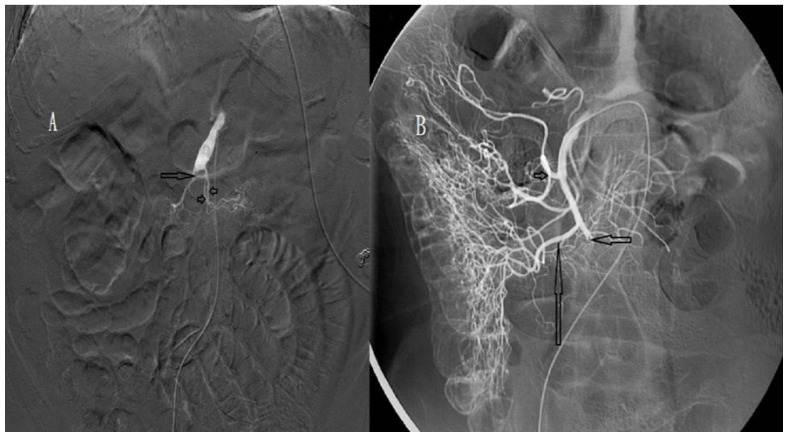
(**A**) Angiography of an 83-year-old woman, showing complete occlusion (arrow) of the superior mesenteric artery (SMA) around the proximal jejunal arteries (arrowheads). Distal main trunk of the SMA is not visible. (**B**) Angiography of an 80-year-old man, showing complete occlusion (short arrow) of the SMA. The middle colic artery (arrowhead) and right colic artery (long arrow) are visible.

**Figure 2 biomedicines-11-00267-f002:**
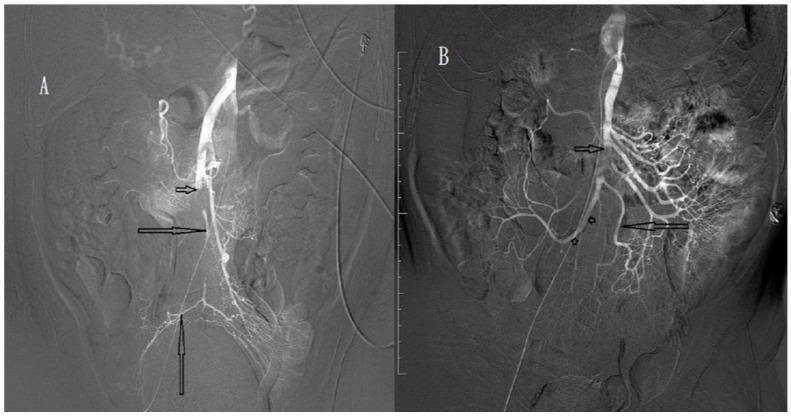
(**A**) Angiography of a 67-year-old man, showing incomplete occlusion (short arrow) of the superior mesenteric artery (SMA). Distal main trunk of the SMA and ileocolic artery (long arrows) are visible. (**B**) Angiography of a 78-year-old man, showing incomplete occlusion (short arrow) of the SMA. The ileal artery (long arrow) and ileocolic artery (arrowheads) are visible.

**Figure 3 biomedicines-11-00267-f003:**
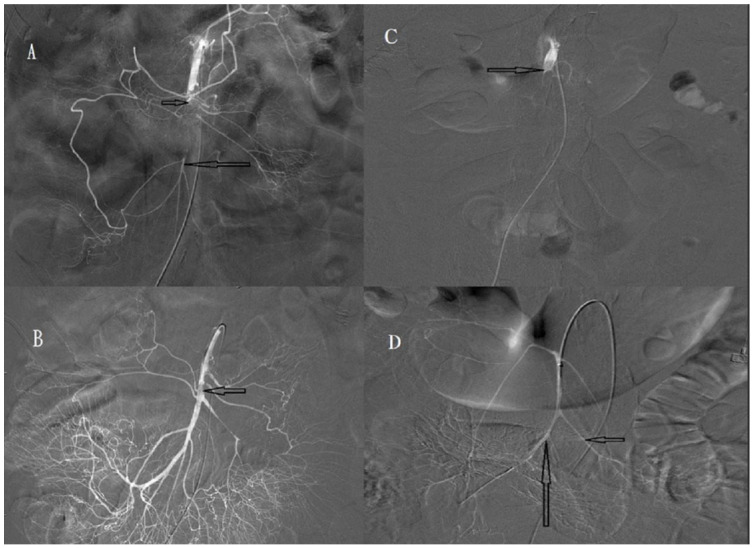
(**A**) Angiography of a 77-year-old man, showing incomplete occlusion (short arrow) of the superior mesenteric artery (SMA). Distal main trunk of the SMA (long arrow) is visible. (**B**) Angiography obtained after one day of urokinase infusion, showing near-total recanalisation of the SMA with appearance of the distal branches. The thrombus decreased in size, but remained (arrow). (**C**) Angiography of a 78-year-old man, showing complete occlusion (arrow) of the superior mesenteric artery (SMA). Distal main trunk of the SMA is not visible. (**D**) Angiography obtained after 26 h of urokinase infusion, showing partial recanalisation of the SMA. The ileal artery (short arrow) and ileocolic artery (long arrow) are visible, but most of the jejunal arteries are not visible.

**Figure 4 biomedicines-11-00267-f004:**
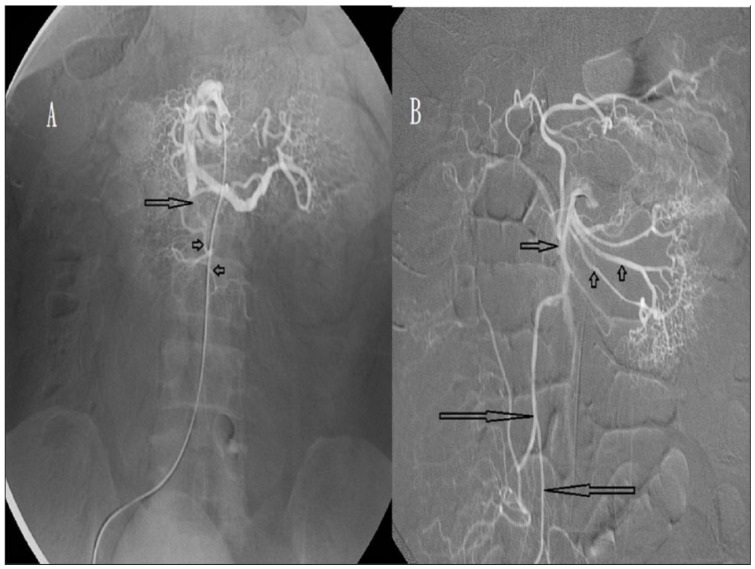
(**A**) Angiography of a 50-year-old man, showing incomplete occlusion (arrow) of the superior mesenteric artery (SMA). Distal main trunk of the SMA is visible (arrowheads). (**B**) Angiography obtained after one day of urokinase infusion, showing partial recanalisation of the previous occlusion (short arrow). Most of the jejunal arteries (arrowheads) and colic arteries (long arrow) are visible.

**Table 1 biomedicines-11-00267-t001:** Demographic data and clinical characteristics of the 13 patients with superior mesenteric artery (SMA) occlusion.

Characteristics	Patients (*n* = 13)
Gender	
Male, *n* (%)	9 (69.2)
Female, *n* (%)	4 (30.8)
Age (years) Median (IQR)	73.0 (24.0)
History	
Hypertension, *n* (%)	11 (84.6)
Atrial fibrillation, *n* (%)	9 (69.2)
Heart disease (CAD, CHF etc.), *n* (%)	6 (46.2)
Diabetes mellitus, *n* (%)	4 (30.8)
CVA, *n* (%)	3 (23.1)
Time from abdominal pain to emergency department (hours) Median (IQR)	9.0 (7.5)
Shock at triage, *n* (%)	0 (0)
Bloody stool, *n* (%)	5 (38.5)
Initial serum WBC (U/L) Median (IQR)	13,200 (8350.0)
Initial serum hemoglobin (g/dL) Median (IQR)	14.0 (4.5)
Initial serum amylase (U/L) Median (IQR)	121.0 (79.0)
Synchronous intra-abdominal organ ischemia on CT,	2 (15.4%)
Time from abdominal pain to urokinase (hours) Median (IQR)	15.0 (6.0)
Duration of urokinase (hours) Median (IQR)	42.0 (45.0)
Site of occlusion from SMA origin (cm) Median (IQR)	5.0 (2.0)
Location of occlusion ^a^	
Proximal, *n* (%)	4 (30.8)
Distal, *n* (%)	9 (69.2)
Degree of SMA occlusion ^b^	
Complete, *n* (%)	7 (53.8)
Incomplete, *n* (%)	6 (46.2)
Degree of SMA recanalization after urokinase	
Total, *n* (%)	0
Near-total, *n* (%)	5
Partial, *n* (%)	8
Surgery, *n* (%)	7 (53.8)
Bowel resection, *n* (%)	6 (46.2)
Partial omentectomy, *n* (%)	1 (7.7)
Laparoscopy ^c^, *n* (%)	4 (30.8)
Repeated surgery, *n* (%)	5 (38.5)
Overall in-hospital mortality rate, *n* (%)	4 (30.8)
Length of stay (days) Median (IQR)	17.0 (34.0)

IQR: interquartile range, CAD: coronary artery disease, CHF: congestive heart failure, CVA: cerebrovascular accident, CT: computed tomography, SMA; superior mesenteric artery. a: located proximal or distal to the middle colic artery; b: complete and incomplete occlusion, which refers to the main trunk of the SMA occlusion without and with distal branches, respectively; c: two patients underwent further laparotomy.

**Table 2 biomedicines-11-00267-t002:** Demographic characteristics, clinical presentations, imaging findings, response to urokinase therapy and outcomes of the 13 patients with superior mesenteric artery (SMA) occlusion.

No.	Age	Sex	History	Bloody Stool	Synchronous Ischemia on CT	SMA Occlusion		Intra-Arterial Urokinase		Laparotomy		Outcome	Hospital Stay (Days)
						To Orifice (cm)	Degree	Time ^a^ (Hours)	Recanalization	Time ^b^ (Hours)	Procedure		
1	50	M	H/T	No	No	D, 5.0	Incomplete	9	Partial	32	Omenectomy	Alive	9
2	80	M	H/T, AF, CVA	Yes	No	D, 8.0	Complete	15	Partial	24	Bowel resection ^c^	Alive	23
3	88	M	H/T, CVA	No	No	D, 5.5	Complete	16	Partial	55	Bowel resection ^c^	Alive	45
4 ^d^	83	F	H/T, AF, CHF	No	kidney	P, 3.5	Complete	19	Partial		No	Dead	1
5 ^e^	77	F	H/T, AF, CAD	Yes	No	D, 4.0	Incomplete	28	Near-total		No	Alive	17
6	68	M	H/T	Yes	No	D, 5.0	Complete	26	Partial	126	Bowel resection ^c^	Dead	65
7 ^e^	67	M	AF, AAA	No	Spleen, kidney	D, 4.5	Incomplete	15	Near-total		No	Alive	7
8	73	F	H/T, CVA, DM	Yes	No	P, 3.5	Complete	21	Partial	47	Bowel resection ^c^	Dead	36
9	78	M	H/T, AF, CAD	No	No	D, 6.0	Incomplete	14	Near-total		No	Alive	6
10	78	M	H/T, AF, DM	No	No	P, 1.5	Complete	15	Partial	60	Bowel resection	Alive	28
11	77	M	H/T, AF, CAD	Yes	No	D, 5.0	Incomplete	12	Near-total		No	Alive	11
12	61	M	H/T, AF	No	No	P, 3.5	Complete	14	Partial	51	Bowel resection ^c^	Dead	49
13	73	F	AF, CAD, DM	No	No	D, 5.5	Incomplete	18.5	Near-total		No	Alive	7

CT: computed tomography, H/T: hypertension, AF: atrial fibrillation, CVA: cerebrovascular accident, CHF: congestive heart failure, DM: diabetes mellitus, CAD: coronary artery disease, AAA: abdominal aorta aneurysm, D: distal to middle colic artery, P: proximal to middle colic artery. a: time from abdominal pain to urokinase; b: time from abdominal pain to surgery; c: underwent repeated surgery; d: died of shock due to delayed surgery; e: underwent laparoscopic examination alone.

**Table 3 biomedicines-11-00267-t003:** Comparison of demographic data and clinical characteristics between the 6 patients with restored bowel perfusion and the 7 patients with unrestored bowel perfusion after urokinase therapy.

Characteristics	Restored Bowel Perfusion	Unrestored Bowel Perfusion	*p*-Value
No. of patients	6 ^a^	7 ^b^	
Gender			>0.999
Male, *n* (%)	4 (66.7)	5 (71.4)	
Female, *n* (%)	2 (33.3)	2 (28.6)	
Age (years) median, IQR	75.0 (15.0)	78.0 (15.0)	0.251
Bloody stool	2/6 (33.3)	3/7 (42.9)	>0.999
Initial serum WBC (U/L) median, IQR	12,850 (7825.0)	14,500 (9900.0)	0.252
Initial serum hemoglobin (g/dL) median, IQR	14.1 (4.5)	13.2 (4.7)	0.830
Synchronous intra-abdominal organ ischemia on CT,	1 (16.7)	1 (14.3)	>0.999
Time from abdominal pain to urokinase (hours) Median (IQR)	14.5 (9.6)	16.0 (6.0)	0.281
Location of SMA occlusion			0.105
Proximal, *n* (%) Distal *n* (%)	0 (0) 6 (100)	4 (57.1) 3 (42.9)	
Degree of SMA occlusion			0.002
Complete, *n* (%)	0 (0)	7 (100)	
Incomplete, *n* (%)	6 (100)	0 (0)	
Degree of SMA recanalization after urokinase			0.012
Near-total Partial	5 (83.3) 1 (16.7)	0 (0) 7( 100)	
Length of stay (days) median, IQR	8.0 (6.0)	36.0 (26.0)	0.032
Mortality, *n* (%)	0 (0)	4 (57.1)	0.0105

IQR: interquartile, CT: computed tomography, SMA: superior mesenteric artery; a: one underwent partial omentectomy; b: including one patient died of shock due to delayed surgery.

## Data Availability

The datasets retrieved and analysed in this study are not publicly available due to patient privacy but are available from the corresponding author on reasonable request.

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
