# Peer review of "Intra-Arterial Urokinase for Acute Superior Mesenteric Artery Occlusion: A Retrospective 12-Year Report of 13 Cases"

_biomedicines, 2023, doi:10.3390/biomedicines11020267_

Round 1

Reviewer 2 Report

Thank you for allowing me to review this work. A few comments:

- How many patients had thrombotic versus embolic etiology?

- Was mechanical thrombectomy considered (Angiojet, Penumbra, Jeti, etc.)? Especially in the "complete" occlusions, this is critical to restore some flow to the bowel, otherwise it will continue to necrose while awaiting for the infused thrombolytic to take action (which may be 2 or 4 or 6 hours before some lumen is established).

- Why urokinase over tPA/alteplase?

- You can report p-values if you wish but I would avoid using words like "statistically significant." You have very low power at best to detect any differences.

- What is your infusion protocol (standard dose? weight based? how soon after initiation of infusion to you return for recheck?)

- How soon after presentation and after infusion initiation was laparotomy performed? That could also affect the outcomes.

- Your study does not support the statement that urokinase might obviate surgery. I would remove this. You did not randomize urokinase patients to surgery and no surgery. Regardless, the standard of care is to still assess the bowel by direct inspection (laparoscopy or laparotomy). Urokinase infusion may serve as a potential replacement for open thrombectomy, that is possible but not directly assessed in this study although suggested.

Round 2

Reviewer 2 Report

Thank you for your revision. A few comments:

- Please include your infusion protocol in the methods section.

- Please include a statement that catheter-directed thrombolysis is not a covered service and therefore unavailable in your setting.

Author Response

To reviewer 2

Thank you for your kindly comments.

Please include your infusion protocol in the methods section.

The infusion protocol will be added in the revised manuscript

- Please include a statement that catheter-directed thrombolysis is not a covered service and therefore unavailable in your setting.

We will add this statement in the “Discussion section” of the revised manuscript.
